# Landslide Segmentation with Deep Learning: Evaluating Model Generalization in Rainfall-Induced Landslides in Brazil

Lucas Pedrosa Soares [1,2], Helen Cristina Dias [2], Guilherme Pereira Bento Garcia [1,2] and Carlos Henrique Grohmann [2,*]

1 Institute of Geosciences, University of São Paulo (IGc-USP), São Paulo 05508-080, Brazil; lpsoares@usp.br (L.P.S.); guilherme.pereira.garcia@usp.br (G.P.B.G.)
2 Spatial Analysis and Modelling Lab (SPAMLab), Institute of Energy and Environment, University of São Paulo (IEE-USP), São Paulo 05508-080, Brazil; helen.dias@usp.br
* Correspondence: guano@usp.br

**Abstract:** Automatic landslide mapping is crucial for a fast response in a disaster scenario and improving landslide susceptibility models. Recent studies highlighted the potential of deep learning methods for automatic landslide segmentation. However, only a few works discuss the generalization capacity of these models to segment landslides in areas that differ from the ones used to train the models. In this study, we evaluated three different locations to assess the generalization capacity of these models in areas with similar and different environmental aspects. The model training consisted of three distinct datasets created with RapidEye satellite images, Normalized Vegetation Index (NDVI), and a digital elevation model (DEM). Here, we show that larger patch sizes ($128 \times 128$ and $256 \times 256$ pixels) favor the detection of landslides in areas similar to the training area, while models trained with smaller patch sizes ($32 \times 32$ and $64 \times 64$ pixels) are better for landslide detection in areas with different environmental aspects. In addition, we found that the NDVI layer helped to balance the model's results and that morphological post-processing operations are efficient for improving the segmentation precision results. Our research highlights the potential of deep learning models for segmenting landslides in different areas and is a starting point for more sophisticated investigations that evaluate model generalization in images from various sensors and resolutions.

**Keywords:** deep learning; landslides; U-Net; automatic segmentation

## 1. Introduction

Landslides are one of the most frequent and destructive natural hazards worldwide. They are responsible for causing infrastructure damages, economic losses, and victims, mainly when it occurs near human habitation [1–3]. In recent years, increased deforestation, unplanned urbanization, climate change, and population growth have enhanced the impact of these events on human lives and infrastructure [4–8]. In 2021, according to the Emergency Event Database (EM-DAT), landslides were classified as the second most costly disaster and caused 40 billion dollars of economic losses in Germany alone and 234 deaths in India [9].

In South America, Brazil concentrates around 40% of all fatal landslides in the continent [2]; several events that occurred in the past few decades in the country led to social and economical losses [10–12]. Therefore, landslide detection studies have been considered critical in remote sensing [5]. However, despite the importance highlighted by many authors, detailed landslide inventories are still scarce [13–15]. Asia/Oceania and Europe lead the publication of studies about landslide inventory construction [16–19]. Nevertheless, several countries, such as Brazil, lack common procedures to recognize landslide features on the landscape [20]. Landslide inventory maps are used to prepare and validate landslide susceptibility models [16,21–24], evaluate risk and vulnerability [25–31], perform geormorphometric (geomorphology) studies [29,32–39], and evaluate landslide events [40]. Limited

and incomplete data may be a source of bias for these studies since model success depends directly on inventory accuracy [41,42].

Landslide inventory maps are usually prepared using high (HR) or very high (VHR) resolution remote sensing imagery [7]. Detection of landslides can be performed manually by aerial image visual interpretation [43–49], semi-automatically, or automatically by using object-based image analysis (OBIA) algorithms [49–51] and pixel-based classification [52,53]. Manual classification of landslides is the prevailing method [14,54,55], but is costly, exhaustive, and time-consuming, almost impracticable for large areas. OBIA is an alternative method for HR and VHR image analysis. The method is based on objects rather than individual pixels [56]. Object-based approaches have two main steps: segmentation and classification [56,57]. Subsequently expert knowledge can be added to the analysis. After segmentation, several object characteristics can be used to classify landslide areas, such as spectral, spatial, hierarchical, textural, and morphological [56]. Pixel-based methods classify each pixel of the image based on its spectral information, ignoring geometric and contextual information, which increases the salt-and-pepper noise in the results [58–60].

In recent years, deep convolutional neural networks (DCNN) have achieved state-of-the-art results in applications such as semantic segmentation, object detection, natural language processing, and speech recognition [61–65]. However, only a few studies have used DCNNs for landslide detection [7].

The recent literature covers topics that evaluate how different architectures affect the model accuracy; the impact of patch size, sampling, and different layers in the results, and the generalization capacity of deep learning models to detect landslides in different areas. Sameen and Pradhan [66] compared residual networks (ResNets) trained with topographical information fused by convolutional networks with topographical data added as additional channels. The models trained with the fused data achieved f1 score and mean intersection over union (mIoU) that were superior by 13% and 12.96% compared to the other models. Ghorbanzadeh et al. [61] compared state-of-the-art machine learning methods and DCNN using RapidEye images and a DEM, with five meters of spatial resolution. The DCNN that used only spectral information and small windows was the best model, achieving 78.26% on the mIoU metric. Yi and Zhang [67] evaluated the LandsNet architecture in two test areas with different environmental characteristics. The results were optimized with morphological operations and the proposed approach yielded an f1 score of 86.89%. Yu et al. [55] used the enhanced vegetation index (EVI), DEM degradation indexes, and a contouring algorithm on Landsat images to sample potential landslide zones with less class imbalance distribution. The trained fully convolutional network (PSPNet) achieved 65% of recall and 55.35% of precision. Prakash et al. [60] used lidar DEM and Sentinel-2 images to compare traditional pixel-based, object-based, and DCNN methods. The deep learning method, U-net with ResNet34 blocks, achieved the best results, with the Matthews correlation coefficient score of 0.495 and the probability of detection rate of 0.72. Prakash et al. [68] evaluated a U-Net in a progressive training with different image spatial resolutions and sensors that used a combination of landslide inventories to predict landslides in different locations around the world. The highest Matthews correlation coefficient achieved was 0.69.

DCNNs, in supervised learning problems, can learn to identify patterns on the training data without the need for complex operations to extract features or pre-processing methods. However, choosing the best network architecture, preparing the training dataset, and tuning the hyperparameters is still a challenge [66,69]. Landslide scar datasets usually have an imbalanced class distribution, with more pixels belonging to background objects, such as urban areas, vegetation, and water, than landslide scars [55]. Therefore, since landslide scars have different shapes and sizes, sampling methods and patch sizes may affect the model accuracy as it can be a way to reduce the class imbalance between the positive and the negative class. Moreover, to the best of our knowledge, only Prakash et al. [68] evaluated the generalization capacity of deep learning models. However, the scenes used to evaluate the models usually are in vegetated areas, where the contrast between the landslide scars

and vegetation allows the models to distinguish the landslides. Moreover, only Yi and Zhang [67] tested post-processing operations to improve the segmentation results.

Thus, the objective of this study is to evaluate model generalization and post-processing techniques with models trained with different datasets and patch sizes in scenes with varying spatial complexity. The main contribution of this paper is as follows:

- Evaluation of model generalization in areas with different scene complexity in Brazil;
- Evaluation of binary opening, closing, dilation, and erosion as post-processing techniques;
- Evaluation of how different patch sizes affect model generalization;
- Evaluation of different datasets on model generalization.

## 2. Study Areas

The study areas (Figure 1) were located in Rio de Janeiro (RJ) and Rio Grande do Sul (RS) states in the southern part of Brazil. The areas located in the city of Nova Friburgo (RJ state) were used to train the deep learning models and were considered as test area 1 (TA1). The area close to the city of Teresópolis, which is also located in RJ state, was used as test area 2 (TA2); and test area 3 (TA3) was located close to the city of Rolante (RS state).

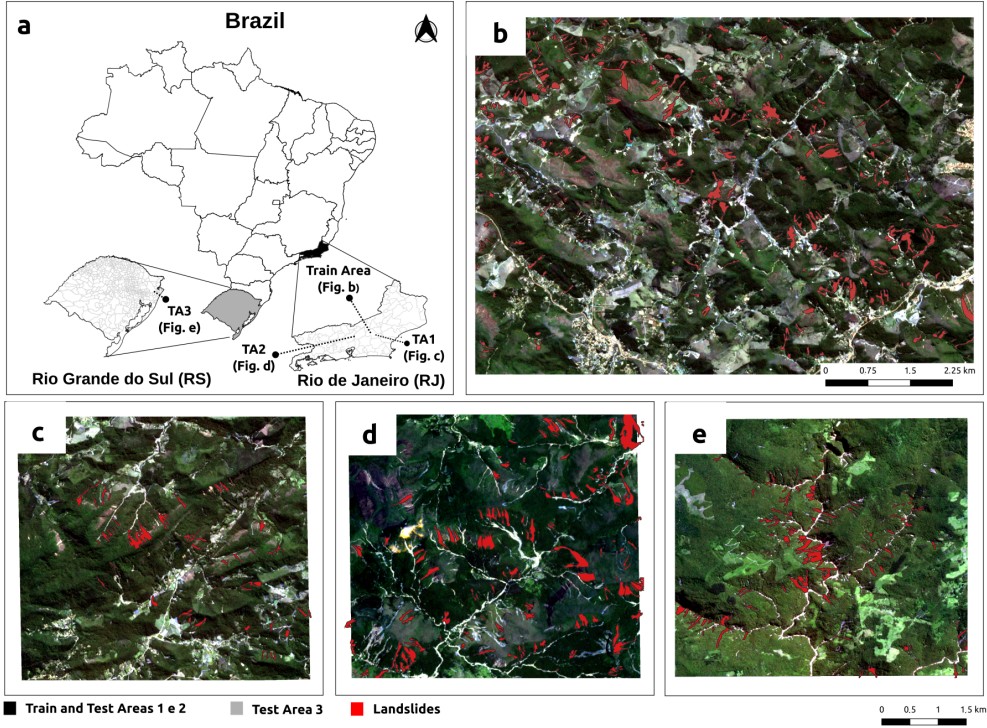

**Figure 1.** Location of the train and test areas used to train and evaluate the deep learning models. (**a**) Location of the train and test areas in Brazil. (**b**) Train Area. (**c**) Test Area 1 (TA1). (**d**) Test Area 2 (TA2). (**e**) Test Area 3 (TA3).

### 2.1. Nova Friburgo and Teresópolis

The mountainous region of Rio de Janeiro encompasses the municipalities of Nova Friburgo, Teresópolis, Petrópolis, Sumidouro, São José do Vale do Rio Preto, and Bom Jardim. In January 2011, an extreme rainfall event (140 mm/h) triggered at least 3500 translational landslides that killed more than 1500 people and disrupted all major city facilities in this mountainous region [11]. This event is considered the worst Brazilian natural disaster [70].

Nova Friburgo and Teresópolis are in the geomorphological unit of Serra dos Orgãos. The geological units have a WSW-ENE trend, and the elevation ranges between 1100 and 2000 m a.s.l. [71]. The geology consists mainly of igneous and metamorphic rocks such as

granites, diorites, gabbros, and gneisses [72] (Figure 2a). According to Köppen's climate classification scheme [73], the climate is subtropical highland (Cwb), with dry winters and mild summers. The annual mean precipitation is 1585.62 mm, with most of the rainfall in November, December, and January [74].

*2.2. Rolante*

The Rolante River Catchment has a drainage area of 828 km$^2$, with altitudes varying from 19 to 997 m a.s.l [75]. The area is inserted in the geomorphological unit of Serra Geral, with a predominance of basaltic rocks and sandstones (Figure 2b). The climate is characterized as very humid subtropical, with precipitation annual average between 1700 and 2000 mm. On 5 January 2017, an extreme precipitation event (272 mm in four hours) triggered at least 300 shallow landslide events in the area [75–78]. The flash flood caused by the material that moved from the slopes into the Mascara river (a tributary of the Rolante River) reached Rolante city.

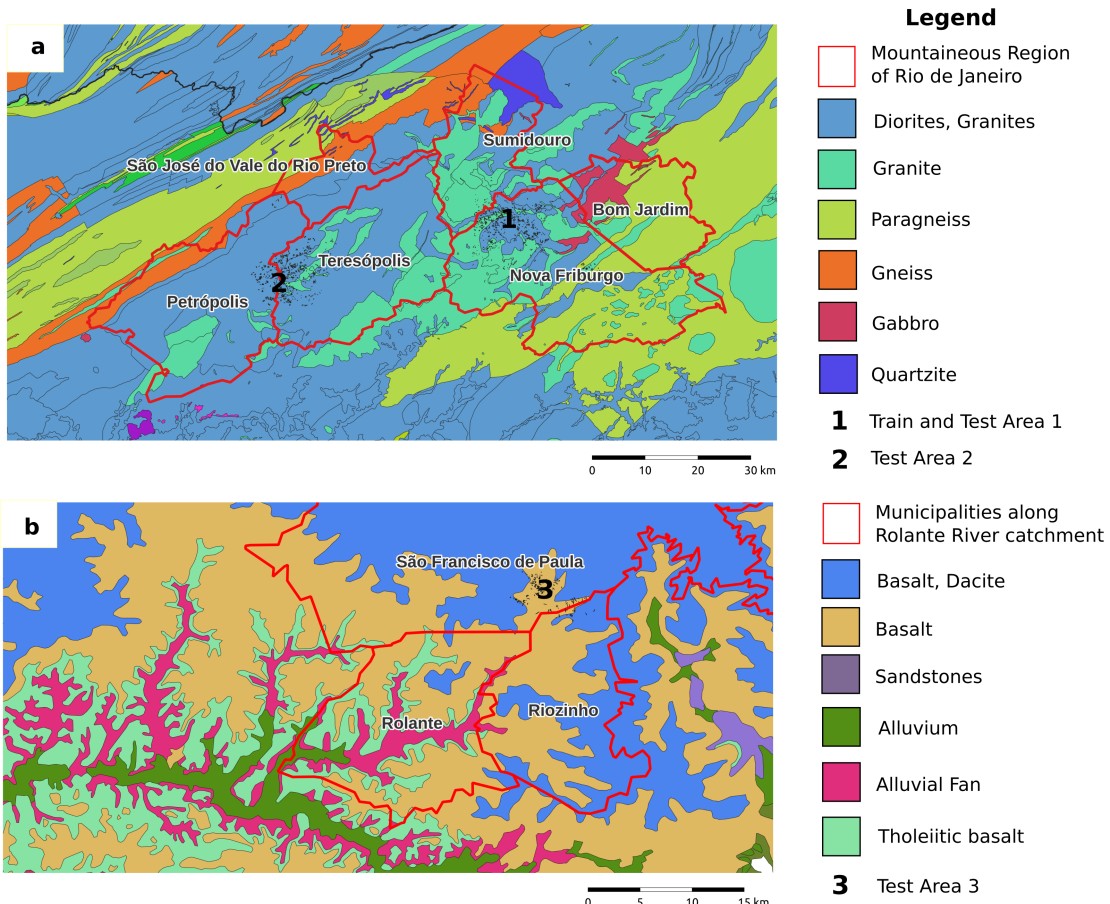

**Figure 2.** Simplified geological maps of the study areas. (**a**) Geological map of the mountainous region of Rio de Janeiro. (**b**) Geological map of the Rolante River area.

**3. Methodology**

The methodology applied in this study consists of four parts: pre-processing, training, evaluation, and post-processing (Figure 3). In the pre-processing step, the data were prepared to serve as the input to the U-Net models. Three different datasets were created to train the models. The sampling was done with regular grids in four different patch sizes: 32 × 32, 64 × 64, 128 × 128, 256 × 256. Augmentation consisted of random rotations, vertical and horizontal flips, and was used to keep the sample size the same among the different patch sizes. The training was done using the Tensorflow 2.0 Python Deep Learning Framework and used grid search to find the optimal hyperparameters. The

evaluation step used precision, recall, f1 score, and mean intersection over union (mIoU) to evaluate the accuracy of the models and their generalization capacity. The models were tested in three different areas with different scene complexities and locations. The post-processing step consisted of evaluating binary opening, closing, dilation, and erosion morphological operations.

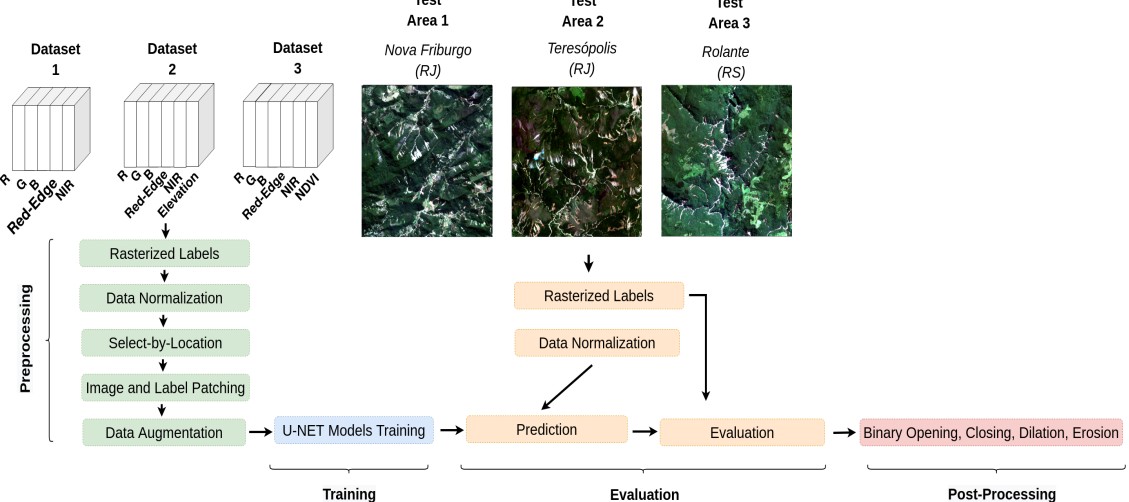

**Figure 3.** Workflow used to prepare the dataset and train, evaluate, and post-process the deep learning models and the results. "R", "G", "B", "Red-Edge", and "NDVI" represent the bands red, green, blue, red-edge, and the normalized vegetation index, respectively.

The data used in this study consist of the spectral information from the RapidEye satellite and topographical data from the Shuttle Radar Topography Mission (SRTM–[79]). RapidEye consists of a constellation of five identical satellites with high-resolution sensors with a 6.5 m nominal ground sampling distance at nadir. The orthorectified products are resampled and provided to users at a pixel size of 5 m. The data are acquired with a temporal resolution of 5 days in five spectral bands: blue (440–510 nm), green (520–590 nm), red (630–685 nm), red-edge (690–730 nm), near-infrared (760–850 nm) [80]. The SRTM acquired interferometric radar data with dual antennas and provided data with 1 arc-second (30 m) spatial resolution. The mission used single-pass interferometry radar to acquire two signals simultaneously by using two different radar antennas. Differences between the two signals permit the calculation of surface elevation [81].

This work used the RapidEye 3A product (orthorectified, radiometric, and geometric corrections) and was acquired from the Planet Explorer website [79]. The acquisition dates of the training and test images are in Table 1. The SRTM product was the 1 arc-second global (30 m).

Three datasets were generated to train and evaluate the deep learning models. All the datasets used the five RapidEye bands. However, dataset 1 used only those five bands, while in dataset 2, the elevation information was added as an extra channel in the image, and in dataset 3, the Normalized Difference Vegetation Index (NDVI) [82] was calculated (Equation (1)) using the red and the near-infrared (NIR) bands and added as an extra channel.

$$\text{NDVI} = \frac{\text{NIR} - \text{RED}}{\text{NIR} + \text{RED}} \tag{1}$$

The landslides were interpreted from the RapidEye and Google Earth Pro version 7.3 imagery and validated with [10,75] to minimize interpretation errors. Table 1 shows the number of landslide polygons interpreted in each scene. Later, the landslides were rasterized using the Rasterio Python library [83] to a binary mask, on which "1" represents

the landslides and "0" the background. The satellite images were normalized to convert all the pixel values into a 0–1 range interval. All the image pixel values were divided by $2^{16}$ (16 bits image). Data normalization helps in model convergence and is a common procedure in the machine learning field.

**Table 1.** Train and test images acquisition date, and the number of landslides present on each scene.

| Images | Acquisition Date (RapidEye/SRTM) | Number of Landslides |
|---|---|---|
| Train Area (Nova Friburgo) | 10 January 2011/23 September 2014 | 455 |
| TA1 (Nova Friburgo) | 10 January 2011/23 September 2014 | 42 |
| TA2 (Teresópolis) | 20 January 2011/23 September 2014 | 117 |
| TA3 (Rolante) | 13 March 2017/23 September 2014 | 110 |

The data were sampled with regular grids in four sizes: $32 \times 32$, $64 \times 64$, $128 \times 128$, $256 \times 256$ pixels. Patching the data in different sizes is an important step to address the differences in the shapes and sizes of the landslides [61]. Moreover, since the patch sizes are directly correlated with the balance between the positive (landslides) and the negative (background) classes, training the models with different sizes is crucial to determine the optimal size for the best model performance in the study areas. A select-by-location operation was used to select only the polygons intersecting landslides. This process ensures that all sampled images will have a small portion of a landslide scar, reducing class imbalance.

Data augmentation allows the use of the annotated data more efficiently during the training phase [67,84]. In this work, because the data were sampled in different patch sizes, the smaller patch sizes have more samples than the larger ones. Hence, comparing the models trained with varying patch sizes may not be fair as the different sample sizes may affect the training of the deep learning models [4]. Thus, to keep the same sample size for all the models, augmentation processes of random rotations and vertical and horizontal flips were performed in the sampled data with patch sizes of $64 \times 64$, $128 \times 128$, and $256 \times 256$ pixels.

*3.1. U-Net*

U-Net [85] is a fully convolutional network developed for the segmentation of biomedical images. This type of architecture does not use fully connected layers in their structure; instead, they have an encoder–decoder architecture with just convolutional layers (Figure 4). The encoder path is responsible for classifying the pixels without taking the spatial location into account, while the decoder path uses up-convolutions and concatenation to recover the spatial location of the classified pixels and return a mask with the same dimensions of the input image.

The convolutional blocks on the encoder path have two $3 \times 3$ convolutional layers, activated with the Rectified Linear Unit (ReLU) function, and followed by a max-pooling operation that reduces the spatial dimension by 2. The dropout layer was used with a 0.5 probability after each max-pooling to randomly deactivate some of the layers of the network as a method to reduce the overfitting.

The convolutional layers are responsible for creating feature maps of the input image to allow the model to predict the landslide. During the training step, the $3 \times 3$ kernels present in these layers are calibrated to find specific features of the landslides. The nonlinear activation function ReLU was calculated according to Equation (2). The use of ReLU increased the degrees of freedom of the computed function, which allows the model to learn nonlinear patterns present in the data [86]. The max-pooling layers with $2 \times 2$ kernels

translate around the image, obtaining only the highest values and reducing the image dimensions by half. This operation is essential to reduce the computation cost and to preserve the values with the highest relevance. Dropout [87] layers are commonly used in the training phase to reduce the complexity of the model and, consequently, the overfitting with random deactivation of the layers with a p probability. In the architecture used in this study, the dropouts were implemented after each max-pooling layer in the encoder path.

On the decoder path, $2 \times 2$ up-sampling operations increase the data's spatial dimension to concatenate feature maps with the same dimension from the encoder path. Then, the concatenated data serve as input for two convolutional layers before another up-sampling operation. At the last layer, a sigmoid function converts the output into a binary mask. The $2 \times 2$ kernels of the transposed convolutions learn how to increase the dimensions of the feature maps during the training step and increase the size of the feature maps by 2. The sigmoid function (Equation (3)) converts the values to the 0–1 range at the last layer.

$$ReLU = max(0, x) \tag{2}$$

$$\sigma(x) = \frac{1}{1 + e^{-x}} \tag{3}$$

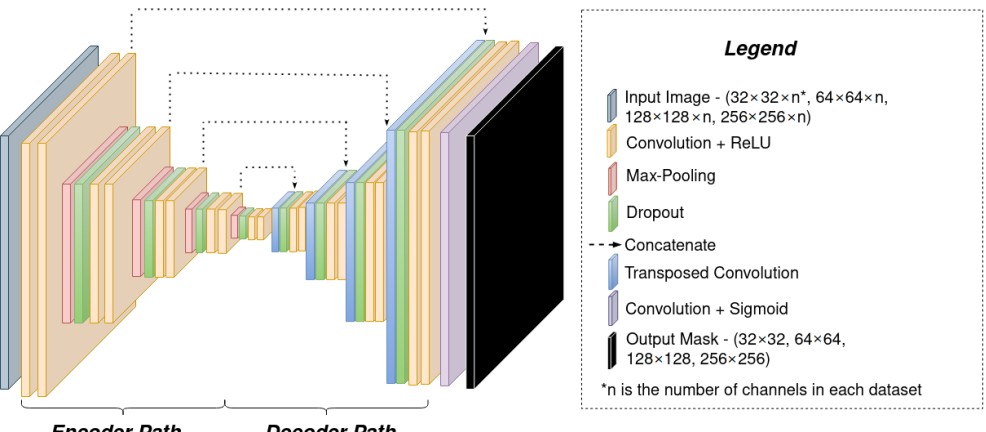

**Figure 4.** U-Net network architecture.

The models were trained for 200 epochs with a dynamic learning rate of 0.001 that reduces by 0.1 in a loss function plateau. Binary Cross Entropy and Adam function were used as the loss and optimization function, respectively. The models were trained with four different batch sizes (16, 32, 64, 128 samples). The model's weights were saved when the validation loss function decreased to reduce the overfitting. The models were trained on Keras [88] and Tensorflow 2.0 [89] Python libraries. Moreover, 30% of each dataset was used as validation data. The training was held in a NVIDIA$^{TM}$ GeForce RTX 2060 GPU (8 GB memory, NVIDIA, Santa Clara, CA, USA).

### 3.2. Validation Metrics

The model's performance was evaluated over two test areas by using the f1 score, recall, precision, and mean intersection over union (mIoU) metrics. These metrics are based on true positives (TP), false positives (FP), and false negatives (FN) [61,90,91]. TP are pixels correctly classified as landslides. FP represents the pixels incorrectly classified as landslides, and FN the pixels incorrectly classified as the background. The models that were trained with DEM and NDVI as an additional channel were evaluated on test areas with an additional DEM and NDVI channel. Precision (Equation (4)) defines how accurate the model is by evaluating how many of the classified areas are landslides. The metric is useful for evaluating the cost of false positives. Recall (Equation (5)) calculates how many of the actual positives are true positives. This metric is suitable to evaluate the cost

associated with false negatives. The f1 score (Equation (6)) combines precision and recall to measure if there is a balance between true positives and false negatives. Mean intersection over union (Equation (7)), also known as the Jaccard Index, computes the overlapping of areas between the ground truth (*A*) and the model prediction (*B*) divided by the union of these areas. Then, the values are averaged for each class. *A* value of 1 (one) represents perfect overlapping, while 0 (zero) represents no overlap.

$$\text{Precision} = \frac{\text{True Positives}}{\text{True Positives} + \text{False Positives}} \tag{4}$$

$$\text{Recall} = \frac{\text{True Positives}}{\text{True Positives} + \text{False Positives}} \tag{5}$$

$$\text{f1-Score} = 2 * \frac{\text{Precision} * \text{Recall}}{\text{Precision} + \text{Recall}} \tag{6}$$

$$\text{mIoU} = \frac{A \cap B}{A \cup B} = \frac{\text{True Positives}}{\text{True Positives} + \text{False Positives} + \text{False Negatives}} \tag{7}$$

### 3.3. Post-Processing

In this study, post-processing morphological operations were used to optimize the results. Binary opening, closing, erosion, and dilation operators were evaluated individually and combined to find the greater improvement (Figure 5). The binary opening helps in removing minor errors that do not represent landslide candidates. Meanwhile, closing, which consists of a dilation followed by erosion, fills the holes inside predicted landslides [67]. Erosion is a mathematical morphological operation that erodes the boundaries of the foreground to shrink the landslide candidates and enlarge the background. Dilation opening helps in removing small noises (i.e., "salt") in the landslide prediction and connects small dark cracks. This tends to open background gaps between the landslides [92]. Several parameters were tested to find the optimal configuration for the post-processing operations. The best structuring element was a $3 \times 3$ square and the interaction was done until the results did not change anymore.

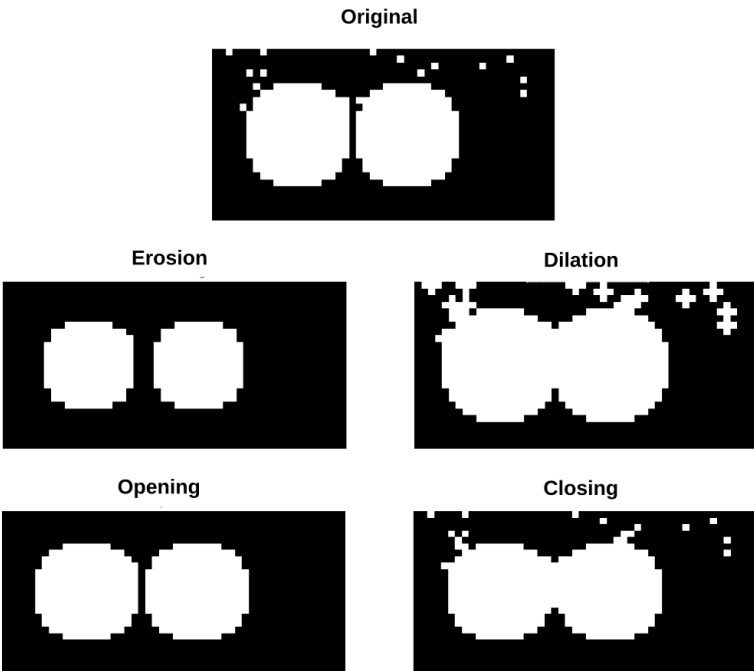

**Figure 5.** Morphological operations used to post-process the segmentation results.

## 4. Results and Discussion

The models were trained with four different patch sizes. In general, models trained with all patch sizes learned the feature maps to detect the landslides. The result shows (Figure 6) that the models trained with 32 × 32 and 64 × 64 pixels achieved the best f1 scores in TA2 (0.53) and TA3 (0.60). In contrast, models trained with 128 × 128 pixels patches achieved the best f1 score results in TA1 (0.52). Since TA1 is located close to the training area, the results show that the models trained with larger patches became better in detecting landslides similar to the training images. This occurs because the patches with greater dimensions facilitate the understanding of the global scene context. Consequently, the deep learning model specialized in detecting landslides with similar spectral and morphological characteristics to the training images. On the other hand, the models trained with the smaller tiles learn the local context of the landslide better. Therefore, they make excessive predictions (low precision), reducing the f1 and mIoU in TA1. However, they achieve better results in TA2 and TA3.

Ghorbanzadeh et al. [61] and Soares et al. [93] evaluated samples with different patch sizes to address the difference in landslide shapes. Ghorbanzadeh et al. [61] conclude that the patch sizes affected the results in a non-systematic way. Meanwhile, Soares et al. [93] observed that models trained with larger patches achieved higher precision and lower recall. Similar results were observed by Prakash et al. [68], where the authors trained the models with 224 × 224 pixels and obtained results with bias towards high precision and lower recall. In this study, the results show that the models tend to achieve better precision and lower recall rates with larger patch sizes. Moreover, comparing the results achieved, it is possible to see that this pattern is more evident in TA1 than the other areas. Thus, once models trained with larger patch sizes become highly specialized in detecting the shape and spectral characteristics of the training area, they tend to achieve better precision in those areas and have worse results in the regions that differ from the training regions.

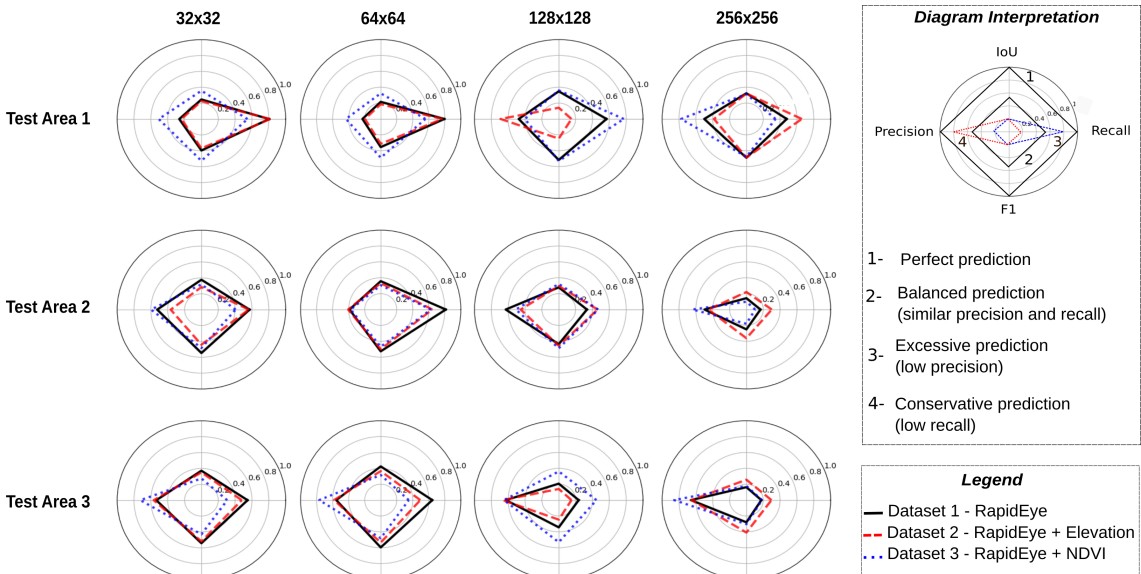

**Figure 6.** F1 score, precision, recall, and mIoU results of the best segmentation models trained with different patch sizes.

Dataset 3 achieved more balanced results than the other datasets, while dataset 1 achieved better f1 and mIoU scores in most models. The higher balance of dataset 3 may be related to the extra NDVI band. NDVI is a band normalization computation with values that range from 0 to 1 and are comparable even in different images. Consequently, it provides information that facilitates the model generalization. Furthermore, dataset 1 has a lower dimensionality (five bands); therefore, according to the Hughes Phenomena [94], it needs less data to train the model than the other datasets with higher dimensionality

(six bands). The topographical data do not improve the results, which are in accordance with the results obtained by Sameen and Pradhan [66]. This may be related to the greater dimensionality of dataset 3 and the SRTM spatial resolution (30 m).

Spectral indices, such as NDVI, are commonly used in remote sensing to help in the interpretation of the spectral signatures of various objects [95]. The correct selection of features based on these indices is crucial in improving traditional machine learning algorithms [96–98]. However, there is a tradeoff between the number of samples and the dimensionality of the data [99]. The extra bands with spectral indices may not improve the algorithm's performance if the dataset is not large enough to overcome the Hughes Phenomena. Moreover, the deep learning convolution operations may learn to calculate the NDVI in the training process from the spectral bands, and the extra band will be redundant. To the best of our knowledge, only the study of Ghorbanzadeh et al. [100] evaluated the impact of using spectral bands and topographic factors (slope, aspect, plan curvature, elevation). However, in this study, the NDVI was used as the basis for landslide detection and was not evaluated; the model architecture used is a classification network that predicts in a pixel-wise manner. For fully convolutional networks, such as U-Net, still, no study evaluates each band's impact on the model performance.

Evaluating the histogram of the best model results in each test area (Figure 7) and it is possible to see that in TA1, the model prediction achieves higher true and false positive rates than the other test areas. Meanwhile, in the other test areas, false negative results were higher. This pattern shows that despite the models' generalization capacity, the areas with different environmental and spectral characteristics from the training area made the model more restrictive. Therefore, fewer landslides are predicted correctly, and the number of false negatives is greater. Prakash et al. [68] observed a similar pattern, where the models trained with different study areas were biased towards high false negatives. Thus, the false positives of TA1 may represent landslides missing in the ground truth inventory, which does not directly represent a poor result.

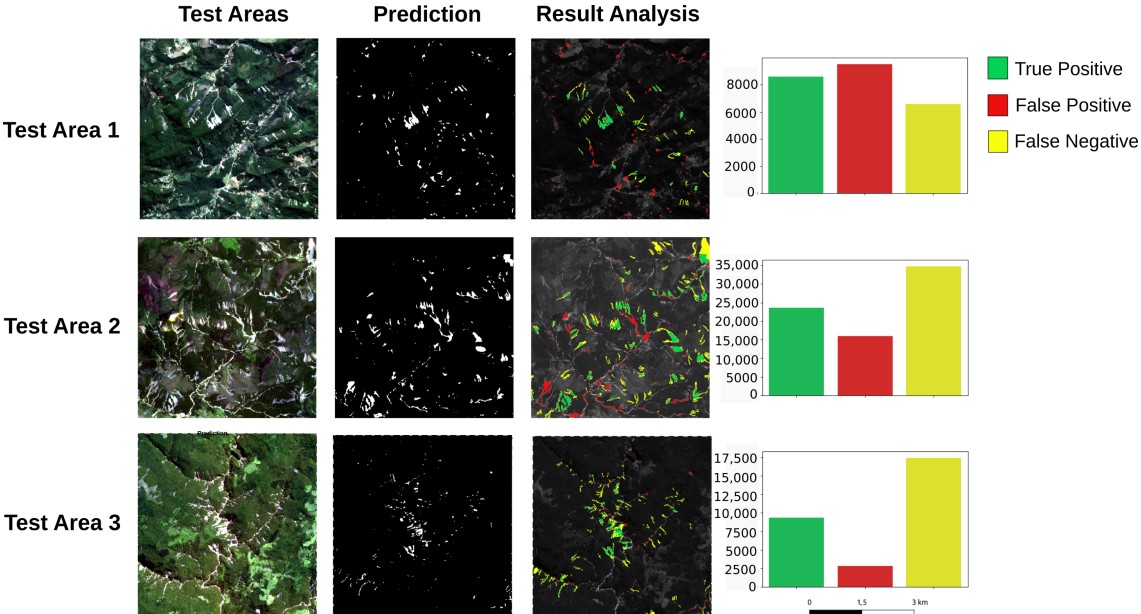

**Figure 7.** Segmentation results of each test area and result histograms showing the number of pixels representing true positives, false positives, and true negatives. These results correspond to the prediction of the model trained with 128 × 128 pixel tiles and dataset 3 for TA1; the model trained with 32 × 32 tile pixels and dataset 1 for TA2, and the model trained with 64 × 64 pixels and dataset 1 TA3.

The complexity of the scene is also an essential factor in evaluating the generalization capacity of the models. In previous studies [61,67,68], the test areas are usually vegetated

areas around the landslide scars. Models trained and tested with these scenes may not be efficient to detect landslides in urban areas due to the higher complexity of the scene and may not be feasible for applications in disaster scenarios. In Qi et al. [98], the authors noticed that the deep learning models had difficulties distinguishing roads and buildings from landslides. In this study, the test areas were chosen to represent areas with different characteristics and complexity. As shown in the histogram of Figure 8, the scores of the models evaluated on TA1 and TA2, which are close to Nova Friburgo and Teresópolis, were reduced by false positives caused by roads and the roofs of the houses. The errors occur in areas with similar spectral responses to the landslides. Since the spatial resolution of the RapidEye images used in this study is 5 m, the model cannot differentiate the shape of the landslides from rivers with increased bedload, areas with bare soil, roads, and roofs. Consequently, the models made these mistakes in all areas. It was expected that the models trained with the DEM layer would overcome the misclassification of the drainage and urban areas since these areas usually have different terrain morphological attributes such as slope and aspect. Probably, these errors occurred due to the coarse resolution of the available DEM (30 m), which cannot clearly detach objects and generalizes the terrain. In Ghorbanzadeh et al. [61], the authors used a 5 m DEM and observed that the DEM helped in differentiating the human settlement areas.

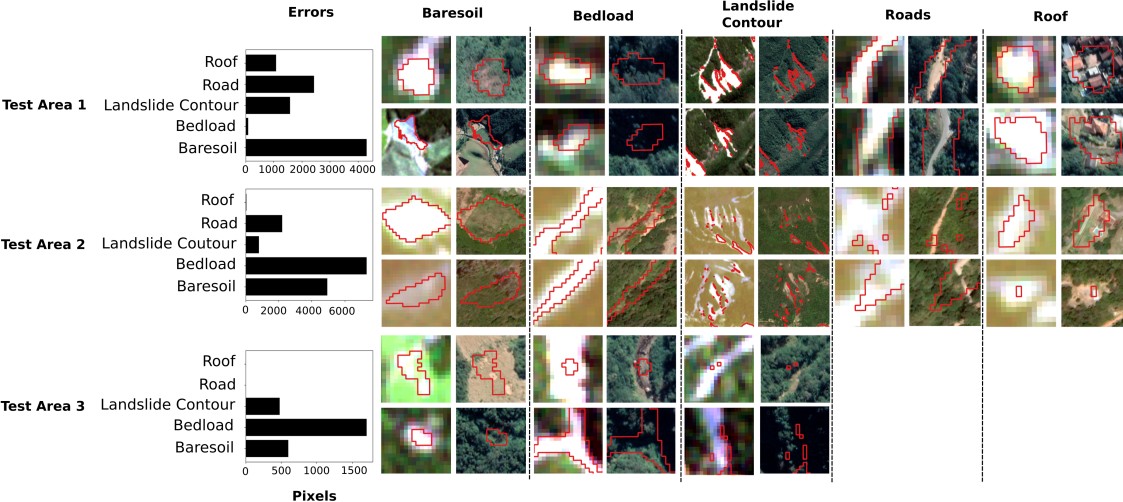

**Figure 8.** Comparison of the most frequent errors made by the deep learning models during landslide segmentation and histograms showing the number of pixels for each error category. Images with lower resolution (**left**) are from the RapidEye satellite, and images with higher resolution (**right**) are from Google Earth.

The post-processing operations were efficient in improving the precision of all test areas. The precision values improved from 0.56 to 0.64 in TA1, 0.57 to 0.65 in TA2, and 0.64 to 0.81 in TA3 (Figure 9). The results of all operations are given in the Supplementary Material Table S1. These results show that the post-processing techniques are efficient in removing the model's systematic errors and are efficient for improving the segmentation precision results.

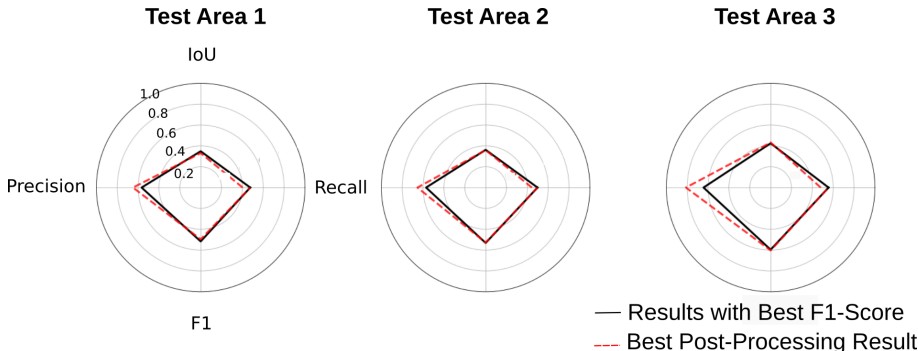

**Figure 9.** F1 score, recall, precision, and mIoU of the best segmentation results of each test area after the post-processing operations. The data used for the post-processing operations correspond to the prediction of the model trained with $128 \times 128$ pixel tiles and dataset 3 for TA1; the model trained with $32 \times 32$ pixel tiles and dataset 1 for TA2, and the model trained with $64 \times 64$ pixel tiles and dataset 1 for TA3.

The morphological operations were evaluated individually and in combination; Table 2 shows the best three combinations' average results for each test area. TA2 and TA3 achieved the best results with the same operations (dilation; closing/dilation; erosion/opening/closing), while in TA1, the operations that yielded the best results were opening; erosion/dilation; and dilation/erosion/opening. Such similarity in the post-processing of TA2 and TA3 and the difference with TA1 may be related to the environmental differences in the training area and the prediction pattern of the model. TA3 achieved better precision results in comparison with the other test areas. This difference seems to be related to each area's landslide characteristics and the model results. Therefore, post-processing operations cannot be generalized and different operations should be tested to find the optimal solution.

**Table 2.** Best post-processing operations and results for each test area. The best results were calculated by summing all the result values. Values in bold represent the best results before the post-processing operations.

| Area | Operation | Recall | Precision | F1-Score | mIoU |
|------|-----------|--------|-----------|----------|------|
| **TA1** | - | **0.57** | **0.47** | **0.52** | **0.35** |
| TA1 | Opening<br>Erosion + Dilation<br>Dilation + Erosion + Opening | 0.48 | 0.56 | 0.52 | 0.35 |
| **TA2** | - | **0.50** | **0.57** | **0.53** | **0.36** |
| TA2 | Dilation<br>Closing + Dilation<br>Erosion + Opening + Closing | 0.44 | 0.65 | 0.53 | 0.36 |
| **TA3** | - | **0.56** | **0.64** | **0.60** | **0.42** |
| TA3 | Dilation<br>Closing + Dilation<br>Erosion + Opening + Closing | 0.48 | 0.81 | 0.60 | 0.43 |

## 5. Conclusions

This study evaluated the generalization capacity of deep learning models and post-processing techniques. The results show that the patch size highly affects the prediction accuracy in areas that are different from the training zone. The larger patch improved the test area results that were close to the training area because larger patches favor a global comprehension of the scene. Consequently, the model becomes specialized in detecting landslides similar to the ones used for the training. On the other hand, the models trained with the smaller patches achieved better results in TA2 and TA3 in locations different from the training zone. This is because the models trained with smaller patches understand the

local context better; they can predict the landslides in a more satisfactory way in different locations. Nevertheless, they also tend to be more restrictive and make more false negative errors. The complexity of the scene is directly correlated with the performance of the models. Therefore, comparing results obtained from different authors, and from different data acquisition methods, such as lidar and Remote Piloted Aircrafts (RPA), may not be reasonable since each training and test area has its own characteristics and complexities. In this way, to better evaluate the machine and deep learning models, a future effort should be made towards an open dataset to evaluate landslide deep learning models. Such open datasets are standard in other computer vision studies such as ImageNet [101], MNIST [102], EuroSat [103] UC Merced Land Use Dataset [104], AID dataset [105], and Brazilian Coffee Scene [106]. Post-processing the results is an efficient step to improve the precision of the segmentation results. The TA3 results improved by 0.17 after combining binary erosion, opening, and closing. The best method to post-process the results will depend on the landslides' characteristics and the model results. Therefore, one should test different combinations and parameters in a semi-supervised way to find the optimal solution. The use of spectral indexes seems to help in balancing the precision and recall of the models and improving model generalization. Since spectral indexes have comparable ranges that facilitate model convergence, the calculation of these indexes is important for predicting landslides in areas with different characteristics from the training areas. Future work should evaluate whether the use of these indexes also facilitates landslide detection in images from different sensors and resolutions.

**Supplementary Materials:** The following supporting information can be downloaded at: https://www.mdpi.com/article/10.3390/rs14092237/, Table S1: Post-processing operation results.

**Author Contributions:** Conceptualization, L.P.S. and C.H.G.; methodology, L.P.S.; software, L.P.S.; validation, L.P.S.; formal analysis, L.P.S., H.C.D. and G.P.B.G.; investigation, L.P.S., H.C.D. and G.P.B.G.; resources, L.P.S.; data curation, L.P.S., H.C.D. and G.P.B.G.; writing—original draft preparation, L.P.S.; writing—review and editing, L.P.S., C.H.G., H.C.D. and G.P.B.G.; visualization, L.P.S.; supervision, C.H.G.; project administration, C.H.G.; funding acquisition, L.P.S. and C.H.G. All authors have read and agreed to the published version of the manuscript.

**Funding:** This study was funded by the Sao Paulo Research Foundation (FAPESP) grants #2019/17555-1, #2016/06628-0, and #2019/26568-0 and by Brazil's National Council of Scientific and Technological Development, CNPq grants #423481/2018-5 and #304413/2018-6. This study was financed in part by CAPES Brasil—Finance Code 001.

**Data Availability Statement:** The code used in this research is available at the following link: https://github.com/SPAMLab/data_sharing/tree/main/Landslide_Segmentation_with_Deep_Learning_Evaluating_Model_Generalization_in_Rainfall-Induced_Landslides_in_Brazil (accessed on 1 May 2021).

**Acknowledgments:** Acknowledgments are extended to the Editor-in-Chief and the anonymous reviewers for their critique and suggestions, which helped to improve this paper.

**Conflicts of Interest:** The authors declare no conflict of interest.

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
