# Peer review of "Landslide Segmentation with Deep Learning: Evaluating Model Generalization in Rainfall-Induced Landslides in Brazil"

_remotesensing, doi:10.3390/rs14092237_

Round 1
Reviewer 1 Report
The minor revisions in the figures are necessary before the final draft.
- Figure 1-A : The characters are too small. Please use the larger characters.
- Line 109 ; (350mm/48h) is an extreme rainfall? If possible, please show the highest intensity per hour.
- Figure 2 ; The characters are too small.
- Line 147 ; “nm” What does this unit stand for?
- Table 1 ; The numbers of Landslides in Train Area are 455, The date include the landslides occurring periodically at the same locations?
- Figure 5 ; The characters are too small.
- Figure 6, 7, 8 ; The characters are too small.
- Figure 7 is the most important figure. It is better to enlarge the image figures themselves.
Reviewer 2 Report
Dear Authors,
I have add my comments to the attached pdf file.
With best regards

Reviewer 3 Report
Summary:
In this manuscript, the authors develop a deep learning model to detect and segment landslide events from the remote sensing images. The study area is in Brazil, but the authors focusses on different climate regimes and elevation within Brazil, thereby evaluating the generalizability and transferability of the model. Developing a generalized model for landslide detection is a challenging problem and the results shown in the manuscript definitely sheds light. This topic is also of interest to the readers of “Remote Sensing”. I have minor suggestions to enhance the readability.
Detailed comments:
- Datasets: The data from RapidEye and SRTM has different spatial resolution. How is this treated with respect to patch size?
- Table 1: How are the number of landslides listed here calculated? Are these pixel-based or object-based? Please mention it.
- Line 164: What do the authors mean by “characteristics of landslides”? Table 1 only has the number of landslides, date and area.
- Figure 2: Include “Red-Edge” also in the caption.
- Figure 3: Model architecture: Not sure if this is correct, but I think the input image size should be three dimensional i.e., the image patch covers 2 dimension and third dimension is different channels (like R, G, B, Red-edge, NIR and DEM/NDVI). Please check this.
- Line 318: “the scene”
- Line 331: What is “crispy resolution”?
- Line 335: The sentence needs paraphrasing. The numbers are can be represented in a better way.
- Figure 6 and 8: Are these results for any particular patch size or the best patch size for the particular test area? Also, what is the dataset used to train the model
- The post-processing operations do improve precision. But, on the other hand, they reduce Recall. Why is having a better precision is more important that better Recall?
- It looks like post processing operations cannot be generalized since TA1 showed better results with different post processing operations compared to TA2 and TA3. Please explain this with more details. This is explained in the conclusion but missing in the results section where Table-2 is discussed.
